# Linear Multistep Solver Distillation for Fast Sampling of Diffusion Models

**Yuchen Liang, Xiangzhong Fang**
School of Mathematical Sciences
Peking University
ycliang@pku.edu.cn

**Hanting Chen, Yunhe Wang**
Huawei Noah's Ark Lab
chenhanting@huawei.com

## Abstract

Sampling from diffusion models can be seen as solving the corresponding probability flow ordinary differential equation (ODE). The solving process requires a significant number of function evaluations (NFE), making it time-consuming. Recently, several solver search frameworks have attempted to find better-performing model-specific solvers. However, predicting the impact of intermediate solving strategies on final sample quality remains challenging, rendering the search process inefficient. In this paper, we propose a novel method for designing solving strategies. We first introduce a unified prediction formula for linear multistep solvers. Subsequently, we present a solver distillation framework, which enables a student solver to mimic the sampling trajectory generated by a teacher solver with more steps. We utilize the mean Euclidean distance between the student and teacher sampling trajectories as a metric, facilitating rapid adjustment and optimization of intermediate solving strategies. The design space of our framework encompasses multiple aspects, including prediction coefficients, time step schedules, and time scaling factors. Our framework has the ability to complete a solver search for Stable-Diffusion in under 12 total GPU hours. Compared to previous reinforcement learning-based search frameworks, our approach achieves over a $10\times$ increase in search efficiency. With just 5 NFE, we achieve FID scores of 3.23 on CIFAR10, 7.16 on ImageNet-64, 5.44 on LSUN-Bedroom, and 12.52 on MS-COCO, resulting in a $2\times$ sampling acceleration ratio compared to handcrafted solvers.

## 1 Introduction

Diffusion models (Sohl-Dickstein et al., 2015; Ho et al., 2020; Song et al., 2021b) have gained widespread success in various applications including image generation (Dhariwal & Nichol, 2021; Rombach et al., 2022), audio synthesis (Kong et al., 2021; Chen et al., 2021), video generation (Ho et al., 2022b;a; Blattmann et al., 2023), and text-to-image synthesis (Saharia et al., 2022; Ruiz et al., 2023; Podell et al., 2024; Esser et al., 2024). When generating samples, diffusion models perform reverse solving of a predefined Stochastic Differential Equation (SDE) or its corresponding Probability Flow Ordinary Differential Equation (ODE) (Song et al., 2021b). This solving process often required hundreds of function evaluations (NFE), making it extremely time-consuming compared to classical generative models like Generative Adversarial Networks (GANs) (Goodfellow et al., 2014).

Fortunately, significant advancements have been made in accelerating the sampling process of diffusion models. Existing acceleration methods can be broadly categorized into two classes. The first class of methods involves an additional distillation training phase (Luhman & Luhman, 2021; Salimans & Ho, 2022; Song et al., 2023; Sauer et al., 2023; Luo et al., 2024). Distilled models require only 1-4 NFE to generate high-quality samples. However, the distillation phase typically requires thousands of GPU hours, posing a significant training cost, especially for large scale models. Additionally, many of distilled models lack the ability to perform image processing tasks, such as image editing and restoration (Yu et al., 2023; He et al., 2023; Wang et al., 2022; Chung et al., 2023; Zhu et al., 2023). We will provide a more detailed discussion in Sec. 2.2.

NFE=5

NFE=10

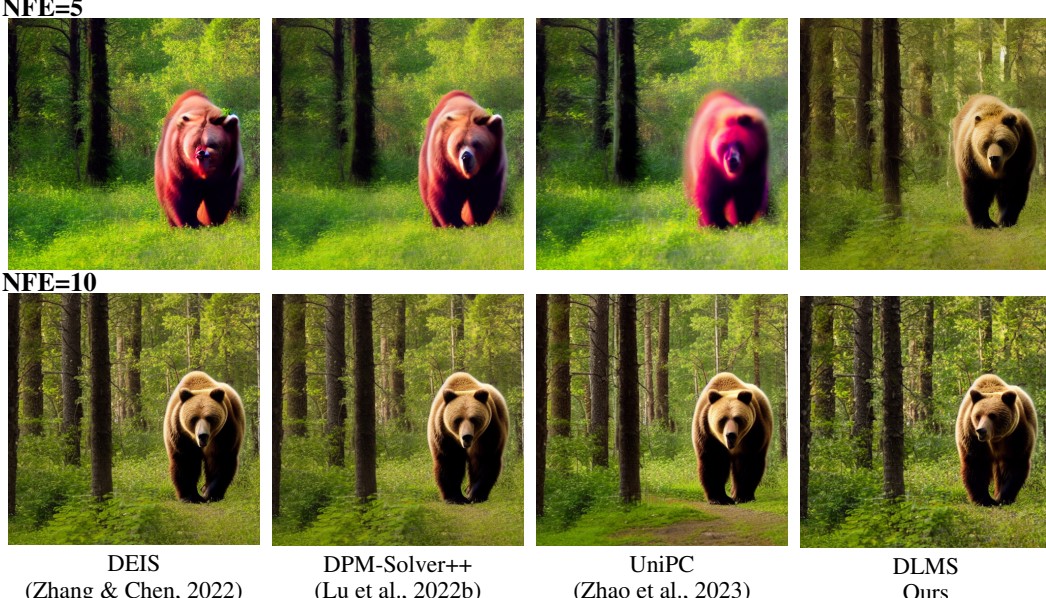

|                      |                         |                      |              |
|----------------------|-------------------------|----------------------|--------------|
| DEIS                 | DPM-Solver++            | UniPC                | DLMS         |
| (Zhang & Chen, 2022) | (Lu et al., 2022b)      | (Zhao et al., 2023)  | Ours         |

Figure 1: Synthesized images of Stable-Diffusion (Rombach et al., 2022) with default classifier-free guidance scale 7.5 and text prompt *"A large **brown** bear walking through a forest"*. Our proposed DLMS can generate more realistic and visually detailed images compared with previous handcrafted samplers (Zhang & Chen, 2022; Lu et al., 2022b; Zhao et al., 2023).

And the second class, which is also widely regarded, focuses on designing efficient solvers without further training the model. Techniques such as parameterization, exponential integrators, and higher-order solvers have successfully reduced the NFE to 15-20 for the sampling process (Lu et al., 2022b; Zhao et al., 2023; Zhang & Chen, 2022; Liu et al., 2022; Lu et al., 2022a; Song et al., 2021a; Karras et al., 2022). However, when the NFE is less than 10, the sample quality deteriorates significantly.

The challenges have given rise to a series of search frameworks seeking more efficient solvers. However, due to the iterative nature of solving processes, it is difficult to predict the impact of intermediate solving strategies on the final sample quality, making the search process difficult and inefficient. These frameworks still often require tens of GPU hours and were limited to specific design space such as time steps (Watson et al., 2022; Li et al., 2023; Liu et al., 2023a; Sabour et al., 2024; Chen et al., 2024), parameterization mode (Zheng et al., 2023), or solver combination (Liu et al., 2023b). Recently, some work (Shaul et al., 2024; Zhang et al., 2024) has attempted to use solver distillation to design solvers, but the performance improvements have been relatively limited.

In this paper, we propose a novel solver distillation framework that greatly accelerates the search efficiency. We begin by presenting a unified prediction formula for linear multistep solvers. Next, we introduce our solver distillation algorithm, which allows a student solver to replicate the sampling trajectory of a teacher solver that uses more steps. We employ the mean Euclidean distance between the student and teacher sampling trajectories as a metric, enabling quick adjustments and optimization of intermediate solving strategies. Compared to previous reinforcement learning-based search frameworks, our approach achieves over a $10\times$ increase in search efficiency.

Our framework has the ability to complete a solver distillation for Stable-Diffusion (Rombach et al., 2022) in less than 1.5h on 8 NVIDIA V100 GPUs. The design scope includes the time steps, the time scaling factors and the prediction coefficients. We extensively evaluate our approach on various resolution datasets in both pixel space and latent space. The *Distilled Linear Multistep Solver* (DLMS) significantly surpasses previous handcrafted and search-based solvers. Compared to handcrafted solvers, DLMS achieves a $2\times$ sampling acceleration ratio.

## 2 BACKGROUND AND RELATED WORK

### 2.1 DIFFUSION MODELS

Given the data distribution $p_0$, diffusion models employ a *forward process* $\mathbf{x}_t = \alpha_t \mathbf{x}_0 + \sigma_t \epsilon, t \in [0, T]$ with marginal distribution $\{p_t\}_0^T$ to gradually degenerate the data $\mathbf{x}_0 \sim p_0$ with Gaussian

noise $\epsilon \sim \mathcal{N}(\mathbf{0}, \boldsymbol{I})$. The *noise schedule* $\alpha_t, \sigma_t > 0$ is designed to make $p_T$ approximately a pure Gaussian distribution $\mathcal{N}(\mathbf{0}, \tilde{\sigma}^2 \boldsymbol{I})$. Notably, there exists a corresponding *Probability Flow* ordinary differential equation (PF-ODE) (Song et al., 2021b):

$$\mathrm{d}\mathbf{x}_t = \left[ f(t)\mathbf{x}_t - \frac{1}{2}g^2(t)\nabla \log p_t(\mathbf{x}_t) \right] \mathrm{d}t, \tag{1}$$

where $f(t) = \frac{\mathrm{d}\log\alpha_t}{\mathrm{d}t}, g^2(t) = \frac{\mathrm{d}\sigma_t^2}{\mathrm{d}t} - 2\frac{\mathrm{d}\log\alpha_t}{\mathrm{d}t}\sigma_t^2$ (Kingma et al., 2021). The PF-ODE shares the same marginal distribution $p_t$ as the forward process. And the score function $\nabla \log p_t(\mathbf{x}_t)$ can be expressed using Tweedie's formula (Robbins, 1992):

$$\nabla \log p_t(\mathbf{x}_t) = \frac{\alpha_t \mathbb{E}[\mathbf{x}_0|\mathbf{x}_t] - \mathbf{x}_t}{\sigma_t^2} = -\frac{\mathbb{E}[\epsilon|\mathbf{x}_t]}{\sigma_t}. \tag{2}$$

To estimate the score function, a neural network $\epsilon_\theta(\mathbf{x}_t, t)$ is trained to predict $\mathbb{E}[\epsilon|\mathbf{x}_t]$ via the least square estimation by minizing the $L_2$ loss

$$\mathbb{E}_{\mathbf{x}_0 \sim p_{\mathrm{data}}, \epsilon \sim \mathcal{N}(\mathbf{0}, \boldsymbol{I})} \|\epsilon_\theta(\mathbf{x}_t, t) - \epsilon\|_2^2 \tag{3}$$

for each $t \in [0, T]$. Consequently, we can sample from diffusion models by solving the empirical PF-ODE:

$$\mathrm{d}\mathbf{x}_t = \left[ f(t)\mathbf{x}_t + \frac{g^2(t)}{2\sigma_t}\epsilon_\theta(\mathbf{x}_t, t) \right] \mathrm{d}t \tag{4}$$

from time $T$ to time 0. Additionally, the conditional sampling can be carried out by guided sampling (Dhariwal & Nichol, 2021; Ho & Salimans, 2021). Remarkably, classifier-free guidance (Ho & Salimans, 2021) defines a guided noise predictor:

$$\tilde{\epsilon}_\theta(\mathbf{x}_t, t, c) = s \cdot \epsilon_\theta(\mathbf{x}_t, t, c) + (1 - s) \cdot \epsilon_\theta(\mathbf{x}_t, t, \emptyset), \tag{5}$$

where $c$ is the condition, $\emptyset$ stand for the unconditional sampling, and $s > 0$ is the guidance scale.

In practice, except for the noise predictor $\epsilon_\theta(\mathbf{x}_t, t)$, diffusion models can also be parameterized as data predictor $\boldsymbol{x}_\theta(\mathbf{x}_t, t)$ to predict $\mathbf{x}_0$ or velocity predictor $\boldsymbol{v}_\theta(\mathbf{x}_t, t)$ to predict $\alpha_t \epsilon - \sigma_t \mathbf{x}_0$, where the parameterizations are theoretically equivalent, but have impact in practice performance (Karras et al., 2022; Hang et al., 2023).

## 2.2 IMAGE PROCESSING WITH PRE-TRAINED DIFFUSION MODELS

Pre-trained diffusion models are versatile tools for a variety of image processing tasks. When aiming to manipulate or restore a reference image $y$, one can simply adjust the sampling score function using Bayes' Rule:

$$\underbrace{\nabla \log p_t(\mathbf{x}_t|\mathbf{y})}_{\text{conditional score}} = \underbrace{\nabla \log p_t(\mathbf{x}_t)}_{\text{score}} + \underbrace{\nabla \log p_t(\mathbf{y}|\mathbf{x}_t)}_{\text{guidance term}} \tag{6}$$

The guidance term can be derived from the gradient of a differentiable loss function. For example, FreeDoM (Yu et al., 2023) uses Gram matrices (Johnson et al., 2016) of the intermediate layers of the CLIP image encoder as the loss function.

Another image processing technique is post-processing. These methods achieve the goal of image processing tasks by projecting the estimated value $\boldsymbol{x}_\theta(\mathbf{x}_t, t)$ at each time step onto a predefined manifold $\mathcal{M}_y$ (He et al., 2023; Wang et al., 2022; Chung et al., 2023; Zhu et al., 2023).

While there are efforts towards distillation work for the guided diffusion model (Meng et al., 2023), the high distillation costs make it impractical to distill for every kind of guidance. In contrast, efficient ODE solvers are much more flexible.

## 2.3 FAST SAMPLING WITH EXPONENTIAL INTEGRATORS

Samplers based on exponential integrator have been found to be more efficient than directly solving the ODE (4). Given an initial value $\mathbf{x}_s$ at time $s$, the ODE solution $\mathbf{x}_t$ can be analytically computed

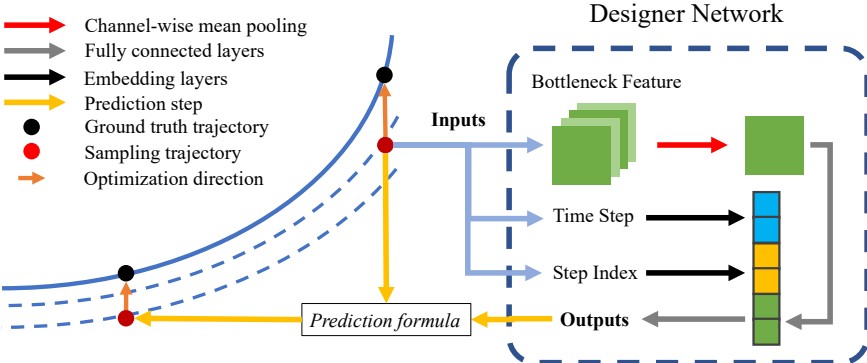

Figure 2: Designer network architecture. We concatenate the embeddings of the bottleneck feature, time step, and step index, and pass them through a fully connected layer to obtain the parameters required for the *prediction formula* (8, 9).

with the following exponentially weighted integral by changing the variable from $t$ to half log-SNR $\lambda_t := \log(\alpha_t/\sigma_t)$ (Lu et al., 2022a;b):

$$\mathbf{x}_t = \frac{\alpha_t}{\alpha_s}\mathbf{x}_s - \alpha_t \int_{\lambda_s}^{\lambda_t} e^\lambda \hat{\epsilon}_\theta(\hat{\mathbf{x}}_\lambda, \lambda)\mathrm{d}\lambda, \quad \mathbf{x}_t = \frac{\sigma_t}{\sigma_s}\mathbf{x}_s + \sigma_t \int_{\lambda_s}^{\lambda_t} e^{-\lambda} \hat{\boldsymbol{x}}_\theta(\hat{\mathbf{x}}_\lambda, \lambda)\mathrm{d}\lambda \tag{7}$$

where $\hat{\epsilon}_\theta(\cdot, \lambda) := \epsilon_\theta(\cdot, t(\lambda)), \hat{\mathbf{x}}_\theta(\cdot, \lambda) := \boldsymbol{x}_\theta(\cdot, t(\lambda))$ and $\hat{\mathbf{x}}_\lambda := \mathbf{x}_{t(\lambda)}$. To predict the integral part, DEIS (Zhang & Chen, 2022) approximates $\epsilon_\theta(\mathbf{x}_t, t)$ with polynomial interpolation w.r.t $t$ , similarly DPM-Solver (Lu et al., 2022a) approximate $\hat{\epsilon}_\theta(\hat{\mathbf{x}}_\lambda, \lambda)$ w.r.t $\lambda$ (equation 7, left) and DPM-Solver++ (Lu et al., 2022b) approximate $\hat{\boldsymbol{x}}_\theta(\hat{\mathbf{x}}_\lambda, \lambda)$ w.r.t $\lambda$ (equation 7, right). UniPC (Zhao et al., 2023) introduces a correcting strategy and various scale functions. And AMED-Solver (Zhou et al., 2024) adopts an approximate mean value method instead polynomial interpolation.

## 2.4 Efficient Sampler Search and Distillation

Recently, several frameworks have incorporated a search phase to enhance the efficiency of samplers. The time step schedule is a commonly explored aspect (Chen et al., 2024; Sabour et al., 2024; Li et al., 2023; Watson et al., 2022; Liu et al., 2023a). Moreover, DPM-Solver-v3 (Zheng et al., 2023) aims to optimize the parameterization mode beyond noise prediction and data prediction. USF (Liu et al., 2023b) strives to identify the most suitable solver for each time step.

Bespoke solver (Shaul et al., 2024) is also a type of solver distillation method, but it only uses the final output to adjust hyperparameters, with a deep and complex computation graph that hinders effective hyperparameter tuning. Zhang et al. (2024) proposed a method that utilizes local mean squared error (MSE) to refine existing solvers, which incurs minimal cost but yields limited effectiveness. We actively use intermediate values in the trajectory and judiciously stop unnecessary gradient backpropagation, greatly enhancing the effectiveness of solver distillation method.

## 3 Methodology

In this section, we present our method to solve the PF-ODE (4). We start with the discussion about the optimal linear multistep prediction strategy. And then we introduce how to achieve the optimal solver with an adaptive time schedule and time scaling factors. Finally, we introduce some practical techniques to further improve performance. A comparison with related work is provided in Appendix A.

## 3.1 Towards Optimal Linear Multistep Solver

Given a time schedule $\{t_n\}_{n=0}^N$ decreasing from $t_0 = T$ to $t_N = 0$, the solvers are trying to predict the ground truth trajectory $\{\mathbf{x}_{t_n}^G\}_{n=0}^N$ with a numerical simulation trajectory $\{\mathbf{x}_{t_n}^S\}_{n=0}^N$ with

$\mathbf{x}_{t_0}^S = \mathbf{x}_{t_0}^G$. Linear multistep solvers aim to predict each local intermediate $\mathbf{x}_{t_n}^G, n \geq 1$ by leveraging the linear combination of $p$ previous outputs from the denoising network. Take the data prediction type (equation 7, right) for example, the *prediction formula* can be expressed as:

$$\boldsymbol{D}_n = \sum_{k=1}^{p} a_k \boldsymbol{x}_\theta(\mathbf{x}_{t_{n-k}}^S, s_{n-k} t_{n-k}), \; \sum_{k=1}^{p} a_k = 1 \tag{8}$$

$$\mathbf{x}_{t_n}^S = \frac{\sigma_{t_n}}{\sigma_{t_{n-1}}} \mathbf{x}_{t_{n-1}}^S - \alpha_{t_n}(e^{\lambda_{t_{n-1}} - \lambda_{t_n}} - 1) \boldsymbol{D}_n \tag{9}$$

where $\{a_k\}_{k=1}^p$ are the prediction coefficients and the time scaling factors $\{s_n\}_{n=0}^N$ are usually set to 1. This implies that the candidate $\mathbf{x}_{t_n}^S$ lies in a $p$-dimensional hyperplane and hence the theoretically optimal $\mathbf{x}_{t_n}^S$ is the linear projection of $\mathbf{x}_{t_n}^G$ onto the hyperplane.

It is worth noting that for each trajectory $\{\mathbf{x}_{t_n}^G\}_{n=0}^N$ and for each step $n$ the optimal coefficients $\{a_k^*\}_{k=1}^p$ are different. Nevertheless, some studies have shown that the ODE trajectories from diffusive models have similar properties(Chen et al., 2024), so it is possible to design a unified $\{a_k\}_{k=1}^p$ that works well on all trajectories just as in most of previous works, but we consider this to be a suboptimal choice.

As mentioned in the previous paragraph, we aim to assign distinct coefficients for each trajectory and step. To achieve this, we establish a mapping $g : (\mathbf{x}_{t_{n-1}}^S, t_{n-1}, n-1) \longmapsto \{a_k\}_{k=1}^p$. Drawing inspiration from Zhou et al. (2024) we utilize the bottleneck feature $\boldsymbol{h}_{t_{n-1}}$ obtained from the pretrained denoising model instead of $\mathbf{x}_{t_{n-1}}^S$ to circumvent additional computational costs. Subsequently, we employ an extremely lightweight *designer network* with approximately 9k parameters, denoted as $g_\phi(\boldsymbol{h}_{t_{n-1}}, t_{n-1}, n-1)$, to generate the coefficients $\{a_k\}_{k=1}^p$. The network architecture is shown in Fig. 2.

Given the availability of the ground truth trajectory $\{\mathbf{x}_{t_n}^G\}_{n=0}^N$, we can establish the optimal solver by minimizing the square distance $d(\mathbf{x}_{t_n}^S, \mathbf{x}_{t_n}^G)$ where the prediction $\mathbf{x}_{t_n}^S$ is computed using equations (8) and (9) with $\{a_k\}_{k=1}^p = g_\phi(\boldsymbol{h}_{t_{n-1}}, t_{n-1}, n-1)$.

In practical terms, we utilize a numerical ground truth trajectory $\mathbf{x}_{t_n}^T$ generated by a teacher solver $\Phi_t$ such as DPM-Solver++ with $M$ interpolation time steps between $t_{n-1}$ and $t_n$, which we denote as $\mathbf{x}_{t_n}^T = \Phi_t(\mathbf{x}_{t_{n-1}}^T, t_{n-1}, t_n, M)$.

## 3.2 Adaptive Time Schedule

Adaptive time schedule is a class of approaches that can significantly improve the efficiency of various solvers. Some previous works (Xia et al., 2024; Zhou et al., 2024) also show that adjusting the time scaling factors $\{s_n\}_{k=1}^N$ properly can improve the accuracy of the solution. In Sec. 3.1, we described how to get the distilled solver for a fixed time schedule $\{t_n\}_{n=0}^N$. In this section, we show how to use our distillation framework to get adaptive time schedules.

When adjusting the prediction coefficients $\{a_k\}_{k=1}^p$ in Sec. 3.1, we chose $\mathbf{x}_{t_{n-1}}^S$ at time $t_{n-1}$ as the starting point to predict $\mathbf{x}_{t_n}^G$. Now, we aim to incorporate $t_{n-1}$ and $s_{n-1}$ into the design space. Therefore, we modify the starting point to be $\mathbf{x}_{t_{n-2}}^S$ at time $t_{n-2}$. To ensure each trajectory has an independent adaptive time schedule rather than a unified schedule, we include the time schedules $\{t_n\}_{n=0}^N$ and $\{s_n\}_{n=0}^N$ in the output of $g_\phi$. This is achieved by setting $\{a_k\}_{k=1}^p, t_{n-1}, s_{n-1} = g_\phi(\boldsymbol{h}_{t_{n-2}}, t_{n-2}, n-2)$ for $n \geq 2$. Let $Q$ denote the buffer list that collects outputs $\boldsymbol{x}_\theta(\mathbf{x}_{t_n}^S, s_n t_n)$. We write the *prediction formula* (8, 9) as:

$$\mathbf{x}_n^S = \Phi(\mathbf{x}_{n-1}^S, t_{n-1}, t_n, \{a_k\}_{k=1}^p, Q). \tag{10}$$

Thus, the distillation phase can be summarized by Algorithm 1. After the distillation phase, Algorithm 2 shows the sampling process with a trained designer network $g_\phi$.

It is worth noting that we have employed the stop gradient operation multiple times in Algorithm 1, which is crucial for the proper functioning of the algorithm. When using the state $\mathbf{x}_{t_{n-2}}^S$ at time $t_{n-2}$ to predict $\mathbf{x}_{t_n}^T$ at time $t_n$, retaining the gradients of $t_{n-2}, t_n, \mathbf{x}_{t_{n-2}}^S, \mathbf{x}_{t_n}^T$ will lead to two issues:

---

**Algorithm 1** Linear Multistep Solver Distillation

---

**Require:** Designer network $g_\phi$, teacher solver $\Phi_t$, number of time steps $N$, number of interpolation time steps $M$, initial timestep $T$, max order $p$.

1: **repeat**
2:     Sample initial value $\mathbf{x}_{t_0}^S = \mathbf{x}_{t_0}^T \sim \mathcal{N}(\mathbf{0}, \tilde{\sigma}^2 \boldsymbol{I})$
3:     $t_0 \leftarrow T, s_0 \leftarrow 1$. Initialize an empty buffer $Q$.
4:     Set the parameter gradient of $\phi$ to 0
5:     $Q \overset{\text{buffer}}{\longleftarrow} \boldsymbol{x}_\theta(\mathbf{x}_{t_0}^S, s_0 t_0)$
6:     Extract the bottleneck feature $h_{t_0}$
7:     **for** $n = 2$ to $N$ **do**
8:         $\{a_k\}_{k=1}^{\max(n-1,p)}, t_{n-1}, s_{n-1} \leftarrow g_\phi(\boldsymbol{h}_{t_{n-2}}, t_{n-2}, n-2)$
9:         $\mathbf{x}_{t_{n-1}}^S \leftarrow \Phi(\mathbf{x}_{n-2}^S, t_{n-2}, t_{n-1}, \{a_k\}_{k=1}^{\max(n-1,p)}, Q)$        $\triangleright$ Generate prediction $\mathbf{x}_{t_{n-1}}^S$.
10:         $Q \overset{\text{buffer}}{\longleftarrow} \boldsymbol{x}_\theta(\mathbf{x}_{t_{n-1}}^S, s_{n-1} t_{n-1})$
11:         Extract the bottleneck feature $h_{t_{n-1}}$
12:         $\{a_k\}_{k=1}^{\max(n,p)}, t_n, s_n \leftarrow g_\phi(\boldsymbol{h}_{t_{n-1}}, t_{n-1}, n-1)$
13:         $t_n \leftarrow \text{sg}(t_n)$        $\triangleright$ Stop the gradient of $t_n$.
14:         $\mathbf{x}_{t_n}^S \leftarrow \Phi(\mathbf{x}_{n-1}^S, t_{n-1}, t_n, \{a_k\}_{k=1}^{\max(n,p)}, Q)$        $\triangleright$ Generate prediction $\mathbf{x}_{t_n}^S$.
15:         **if** n=2 **then**
16:             $\mathbf{x}_{t_{n-1}}^T \leftarrow \text{sg}(\Phi_t(\mathbf{x}_{t_{n-2}}^T, t_{n-2}, t_{n-1}, M))$
17:         **end if**
18:         $\mathbf{x}_{t_n}^T \leftarrow \text{sg}(\Phi_t(\mathbf{x}_{t_{n-1}}^T, t_{n-1}, t_n, M))$        $\triangleright$ Solve the numerical ground truth $\mathbf{x}_{t_n}^T$.
19:         $\mathcal{L}_n(\phi) \leftarrow d(\mathbf{x}_{t_n}^S, \mathbf{x}_{t_n}^T)$        $\triangleright$ Calculate the prediction error at time $t_n$.
20:         Perform backpropagation for $\mathcal{L}_n(\phi)$.
21:         $Q \leftarrow \text{sg}(Q), t_{n-1} \leftarrow \text{sg}(t_{n-1})$        $\triangleright$ Stop the gradient of buffers and $t_{n-1}$
22:     **end for**
23:     Update the parameter $\phi$
24: **until** convergence

---

**Algorithm 2** Distilled Solver Sampling

---

**Require:** Designer network $g_\phi$, number of time steps $N$, initial timestep $T$, max order $p$.

1: Sample initial value $\mathbf{x}_{t_0}^S \sim \mathcal{N}(\mathbf{0}, \tilde{\sigma}^2 \boldsymbol{I})$
2: $t_0 \leftarrow T, s_0 \leftarrow 1$. Initialize an empty buffer $Q$.
3: **for** $n = 1$ to $N$ **do**
4:     $Q \overset{\text{buffer}}{\longleftarrow} \boldsymbol{x}_\theta(\mathbf{x}_{t_{n-1}}^S, s_{n-1} t_{n-1})$
5:     Extract the bottleneck feature $h_{t_{n-1}}$
6:     $\{a_k\}_{k=1}^{\max(n,p)}, t_n, s_n \leftarrow g_\phi(\boldsymbol{h}_{t_{n-1}}, t_{n-1}, n-1)$
7:     $\mathbf{x}_{t_n}^S \leftarrow \Phi(\mathbf{x}_{n-1}^S, t_{n-1}, t_n, \{a_k\}_{k=1}^{\max(n,p)}, Q)$
8: **end for**
9: **return** $\mathbf{x}_{t_N}^S$

---

- Firstly, the gradients from past time steps will not be removed from the computation graph, resulting in a linear increase in memory usage with each step $n$.

- Secondly, $t_n$ will move towards $t_{n-2}$ to ease the prediction difficulty, ultimately resulting in the collapse of the entire time schedule. This is not an ideal outcome.

In addition, we find it necessary to adopt data prediction type (equation 7, right). We believe this is because the data prediction type can automatically align the noise intensity with the time step $t$, thus stabilizing the prediction error for the solvers.

### 3.3 PRACTICAL TECHNIQUES

In this section, we introduce several practical techniques to further improve the performance of DLMS.

**High-order initialization.** Since the prediction formula (8, 9) covers almost all linear multi-step solvers with exponential integrators such as DDIM (Song et al., 2021a), PLMS (iPNDM) (Zhang & Chen, 2022; Liu et al., 2022), DPM-Solver++ (Lu et al., 2022b), UniP-$p$ (Zhao et al., 2023) and DEIS (Zhang & Chen, 2022). We can initialize with these pre-designed solvers by setting the output biases of the designer network $g_\phi$. Therefore, DLMS has a theoretical accuracy that is at least on par with the aforementioned methods. Thus, we still refer to the number of history outputs $p$ used in each step as the "order". Although our framework still works with DDIM (Song et al., 2021a) initialization, using higher-order solvers allows the distillation phase to be completed more quickly with fewer generated trajectories. In our experiments, initializing with any higher-order methods did not show significant differences; for simplicity, we recommend PLMS (Zhang & Chen, 2022; Liu et al., 2022) as the initial solving strategy.

**Analytical First Step (AFS).** In Algorithm 1, the initial time step $t_0 = T$ and time scaling factor $s_0 = 1$ are excluded from the design space. Instead, we can treat $t_0 = T$ as a virtual initial step. Specifically, we set $\boldsymbol{x}_\theta(\mathbf{x}_{t_0}^S, s_0 t_0) = \mathbf{0}$ and $h_{t_0} = \mathbf{0}$, without employing the denoising network. This approach ensures that $t_1$ is the actual first time step at which the diffusion model is engaged, while $t_1$ and $s_1$ remain in the design space. This strategy is fundamentally equivalent to the *analytical first step* (AFS) proposed by Dockhorn et al. (2022).

**Exponential Moving Average (EMA).** Due to NFE limitations, the student solver cannot fully replicate the sampling trajectory generated by the teacher solver. This leads to a non-zero stochastic gradient even at convergence. To reduce the impact of parameter oscillations on solver performance, we introduce EMA updates inspired by diffusion models. During solver distillation, we apply EMA updates with half-lives of 1, 2, and 3 kimg, selecting the best-performing configuration from four parameter sets.

**Inception distance at the final step.** Some search-based frameworks directly use Fréchet Inception Distance (FID) (Heusel et al., 2017) as the optimization objective (Liu et al., 2023b; Watson et al., 2022; Li et al., 2023). In contrast, our method employs square distance as the optimization target. Since square distance is less sensitive to high-frequency information, we observe that optimizing square distance does not always lead to better FID scores. Therefore, in experiments conducted in pixel space, we replace the final step's objective from pixel square distance to Inception Distance (the square distance of features from the Inception network). We only make this replacement in the final step because we find that Inception Distance does not effectively capture the differences between noisy images. In latent space tasks, the issue of high-frequency information is not present. To avoid unnecessary calls to the network modules, we do not perform this replacement in those cases.

# 4 EXPERIMENTS

In this section, we demonstrate that DLMS exhibits significant advantages in both unconditional and conditional sampling with pixel-space and latent-space diffusion models. We conducted experiments across multiple datasets with resolutions ranging from 32 to 512 and compared our approach with current state-of-the-art both handcrafted and search-based solvers. Then, we showcase the benefits from adaptive time steps and time scales, as well as ablation studies of the practical techniques we provided. Finally, we visualize the adaptive time schedules and the samples generated by DLMS.

## 4.1 SETTINGS

In our experiments, we uniformly use the noise schedule $\alpha_t = 1, \sigma_t = t$ from Karras et al. (2022). We initialized the prediction coefficients with PLMS (Zhang & Chen, 2022; Liu et al., 2022), using a uniform time schedule (Ho et al., 2020) and time scaling factors of 1. We use DPM-Solver++ (Lu et al., 2022b) to generate ground truth trajectories. The designer network $g_\phi$ consists of a two-layer MLP with a total parameter count of only 9k. We use Adam as the optimizer with a learning rate of $5 \times 10^{-3}$. To ensure fairness, we use the same random seed to evaluate the FID score. For details, see Appendix B.

## 4.2 MAIN RESULTS

We select the current state-of-the-art artificially designed solvers as baseline methods, including DEIS Zhang & Chen (2022), DPM-Solver++ (Lu et al., 2022b), and UniPC (Zhao et al., 2023). All results are obtained from an open-source toolbox[1], utilizing the recommended settings from the original papers. Detailed results can be found in the Appendix C.

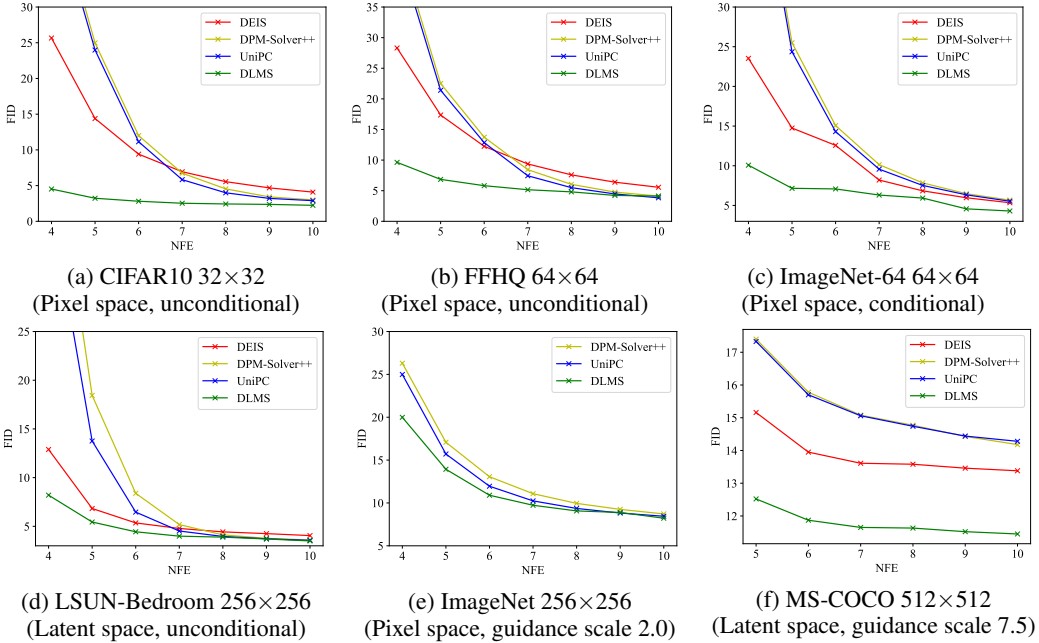

(a) CIFAR10 32×32
(Pixel space, unconditional)

(b) FFHQ 64×64
(Pixel space, unconditional)

(c) ImageNet-64 64×64
(Pixel space, conditional)

(d) LSUN-Bedroom 256×256
(Latent space, unconditional)

(e) ImageNet 256×256
(Pixel space, guidance scale 2.0)

(f) MS-COCO 512×512
(Latent space, guidance scale 7.5)

Figure 3: Comparison of FID↓ scores between DLMS and handcrafted solvers. DLMS demonstrates a significant advantage over handcrafted methods.

Fig. 3 presents a comparison of FID scores between DLMS and previous handcrafted methods. As shown in Fig. 3, handcrafted solvers based on polynomial interpolation exhibit a sharp decline in performance as NFE decreases. In contrast, our proposed DLMS maintains high-quality sampling, demonstrating a significant advantage over baseline methods. In text-to-image generation with Stable-Diffusion (Rombach et al., 2022), DLMS achieves an FID of 12.52 with only 5 NFEs, while the baseline methods require more than 10 NFEs for comparable performance, resulting in a 2× acceleration ratio.

We further compare our method with other search frameworks. The methods include in the comparison are the reinforcement learning-based framework USF (Liu et al., 2023b), the parameterization-focused DPM-Solver-v3 (Zheng et al., 2023). Additionally, we include GITS (Chen et al., 2024), an outstanding adaptive time scheduling method, and AMED-Plugin (Zhou et al., 2024), a closely related work. As shown in Tab. 1, DLMS still stands out among various search frameworks.

Table 1: Comparison of FID↓ on CIFAR10 between DLMS and search-based solvers.

| Solver | NFE | | | | | | |
|---|---|---|---|---|---|---|---|
| | 4 | 5 | 6 | 7 | 8 | 9 | 10 |
| DPM-Solver-v3 | - | 12.21 | 8.56 | - | 3.50 | - | 2.51 |
| USF | 11.50 | 6.86 | 5.18 | 3.81 | 3.41 | 3.02 | 2.69 |
| AMED-Plugin | - | 6.61 | - | 3.65 | - | 2.63 | - |
| GITS | 10.11 | 6.77 | 4.29 | 3.43 | 2.70 | 2.42 | 2.28 |
| **DLMS** | **4.52** | **3.23** | **2.81** | **2.53** | **2.43** | **2.37** | **2.24** |

---

[1] https://github.com/zju-pi/diff-sampler.

In Tab. 2, we compare the total GPU hours required for a 7 NFE solver with USF and DPM-Solver-v3. Due to limitations in code availability, the reported time costs are sourced from original papers and measured on different devices. Nevertheless, our method demonstrates a significant order-of-magnitude advantage, achieving a $10\times$ increase in search efficiency.

Table 2: Comparison of total GPU hours for search phase. DLMS demonstrates a significant order-of-magnitude advantage.

| Solver | CIFAR10 | MS-COCO | Device |
|---|---|---|---|
| DPM-Solver-v3 | 28 | 88 | NVIDIA A40 |
| USF | 12.15 | 106.64 | NVIDIA 3090/A100 |
| **DLMS** | **0.7** | **10** | NVIDIA V100 |

## 4.3 ABLATION STUDY

Tab. 3 demonstrates the ablation effects of each component in the DLMS framework. As shown in the results, the time step-related time schedule and time scaling contribute the most significant improvements.

Table 3: FID results of ablation study on CIFAR10. The time step-related time schedule and time scaling contribute the most significant improvements.

| NFE | 4 | 6 | 8 | 10 |
|---|---|---|---|---|
| **DLMS** | **4.52** | **2.81** | 2.43 | **2.24** |
| w/o AFS | 6.48 | 3.30 | **2.42** | 2.30 |
| w/o bottleneck feature | 4.71 | 3.40 | 2.46 | 2.25 |
| w/o high-order initialization | 4.92 | 3.25 | 2.94 | 2.44 |
| w/o Inception distance | 6.67 | 3.77 | 3.10 | 2.80 |
| w/o time scaling | 7.75 | 3.86 | 3.07 | 2.41 |
| w/o adaptive time schedule | 10.41 | 6.18 | 3.17 | 3.03 |
| Handcrafted (best) | 25.66 | 9.40 | 3.99 | 2.89 |

Another finding is that DLMS performance does not always improve as the number of interpolation time steps $M$ increases. In fact, the fitting capacity of student solvers is limited. When the teacher solver is significantly superior to the student, side effects can arise due to fitting difficulties. Especially for models in the latent space, setting $M = 1$, meaning the teacher has about twice NFEs of the student, yields the best results. The table below illustrates the effects of increasing $M$ on the LSUN-Bedroom dataset.

Table 4: The sensitivity of $M$ on LSUN-Bedroom. Setting $M = 1$ yields the best results on latent diffusion models.

| NFE | 4 | 6 | 8 | 10 |
|---|---|---|---|---|
| **M=1** | **8.20** | **4.44** | **3.89** | **3.50** |
| M=2 | 8.86 | 4.56 | 4.51 | 3.69 |
| M=3 | 11.07 | 4.88 | 4.54 | 3.81 |

## 4.4 VISUALIZATIONS

**Visual Quality.** We present qualitative comparisons in Fig. 1. Handcrafted solvers struggle to generate vegetation and accurately colored bears with 5 NFEs. In contrast, DLMS effectively learns the generation results of the teacher solver with double NFEs. With 10 NFEs, DLMS is the only method that successfully produces the correct lighting and head pose. Additional samples are provided in Appendix D.

**Visualization of adaptive time schedule.** Fig. 4 shows the adaptive time schedules (average of different trajectories) obtained from DLMS with 10 NFE using AFS. By comparing these schedules with handcrafted time schedules such as logSNR (Zheng et al., 2023; Lu et al., 2022b), polynomial (Karras et al., 2022), time uniform (Ho et al., 2020) and time square (Zhang & Chen, 2022), we uncover some intriguing findings. The adaptive time schedules exhibit striking differences across various datasets. For the CIFAR10 and FFHQ datasets, the adaptive time schedules are similar to the time square schedule. In contrast, for ImageNet-64 and LSUN-Bedroom, they resemble the time uniform schedule. Notably, the schedule learned for the Stable-Diffusion model aligns closely with the uniform logSNR schedule, except for the final step.

The observed differences may arise from several factors, including the solving strategy, image resolution, solving space, and guidance. For instance, in the EDM context, DPM-Solver++ (Lu et al., 2022b) and UniPC (Zhao et al., 2023) perform better with logSNR, while DEIS (Zhang & Chen, 2022) demonstrates superior performance with the time square schedule. However, on other models, all three solvers tend to prefer the time uniform schedule. This highlights the importance of simultaneously searching for both time schedules and solving strategies within our framework across different datasets.

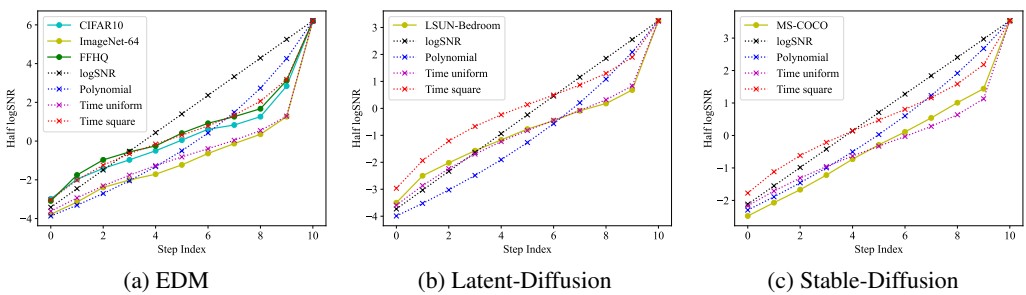

|          |                |                      |
| :------: | :------------: | :------------------: |
| (a) EDM  | (b) Latent-Diffusion | (c) Stable-Diffusion |

Figure 4: Visualization of adaptive time schedule. We use dotted lines to represent handcrated time schedules, and use solid lines to represent adaptive time schedules learned on each data set. DLMS learns different adaptive schedules on different data sets.

## 5 CONCLUSION

We propose a linear multistep solver distillation framework. Our framework enables the student solver to replicate the sampling trajectory of a teacher solver that utilizes more steps, facilitating rapid adjustments and optimization of prediction coefficients, time step schedules, and time scaling factors. Experiments demonstrate the effectiveness of our framework across various resolution datasets, using both pixel-space and latent-space pre-trained diffusion models, and reveal a significant improvement in sample quality with 4∼10 NFEs.

**Limitations and Future Work.** Our framework is currently limited to ODE solvers, while in practice, stochastic samplers (Xue et al., 2024) often outperform deterministic samplers. Therefore, extending our method to stochastic samplers is a promising direction. Additionally, integrating our work with approaches such as Deepcache (Ma et al., 2024) and FreeU (Si et al., 2024) is also worth exploring.

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

## A    COMPARING WITH RELATED METHODS

**Comparing with Progressive Distillation.** Progressive distillation (Salimans & Ho, 2022) is a distillation method for diffusion models that gradually distills the results of multi-step solvers into fewer steps. In our framework, we also employ a multi-step teacher solver and aim to distill the results into our student solver with fewer steps. However, our approach does not require training the parameters of the diffusion model, while the parameter requirements and training duration for progressive distillation (PD) are thousands of times greater than those of our framework.

**Comparing with AMED-Plugin.** AMED-Plugin (Zhou et al., 2024) is a method for selecting intermediate time steps for existing solvers and time schedules. The designer network $g_\phi$ used in our work is modified from AMED-Plugin. In contrast, the DLMS in this paper do not rely on existing solvers or time schedules. AMED-Plugin can be seen as adjusting half of the time schedule, while our proposed adaptive time schedule method is for full time schedule adjustment.

**Comparing with USF.** USF (Liu et al., 2023b) is a search framework based on reinforcement learning. The authors train a predictor network to estimate the FID performance of a solver with specific hyperparameters, using this to guide an evolutionary search process for hyperparameter optimization. The relationship between hyperparameters and performance is complex, making the training of a predictor network both challenging and time-consuming.

In contrast, our framework leverages local prediction error to design an optimal solution strategy, which aligns more closely with the iterative nature of the sampling process and the criteria for manually designing solvers. Furthermore, while USF simply combines existing handcrafted solvers, our approach utilizes prediction formulas (8, 9) to achieve a true unification of multiple solving strategies. This enables us to develop new solving strategies by learning the prediction coefficients $\{a_k\}_{k=1}^p$.

**Comparing with Bespoke solver.** Bespoke solver (Shaul et al., 2024) is also a type of solver distillation method, but it only uses the final output to adjust hyperparameters, with a deep and complex computation graph that hinders effective hyperparameter tuning. We actively use intermediate values in the trajectory and judiciously stop unnecessary gradient backpropagation, greatly enhancing the effectiveness of solver distillation method. For EDM trained on CIFAR10, our method achieved an FID of 2.53 with 7 NFEs, outperforming their result of 2.75 with 20 NFEs.

## B    DETAILED SETTINGS

**EDM on CIFAR10, FFHQ, ImageNet-64.** The sampling on CIFAR10 (Krizhevsky et al., 2009) 32×32 , FFHQ (Karras et al., 2019) 64×64, ImageNet-64 (Deng et al., 2009) 64×64 is based on the pretrained pixel-space diffusion model provided by EDM (Karras et al., 2022). Among these, ImageNet-64 is for conditional sampling and CIFAR10, FFHQ are for unconditional Sampling. The order $p$ for student solver DLMS is set to 4. The number of interpolation time steps $M$ is set to 4. For each dataset, we conduct solver distillation on 20k trajectories. The distillation times are approximately 40×NFE seconds, 80×NFE seconds, and 150×NFE seconds, respectively, on 8 NVIDIA V100 GPUs. We measure sample quality using the FID score calculated on 50k generated images.

**Latent-Diffusion on LSUN-Bedroom.** The unconditional sampling on LSUN-Bedroom (Yu et al., 2015) 256×256, is based on the pretrained latent-space diffusion model provided by Latent-Diffusion (Rombach et al., 2022). The order $p$ for student solver DLMS is set to 3. The number of interpolation time steps $M$ is set to 1. We conduct solver distillation on 10k trajectories. The distillation times are approximately 3×NFE mins on 8 NVIDIA V100 GPUs. We measure sample quality using the FID score calculated on 50k generated images.

**Guided-Diffusion on ImageNet.** The conditional sampling on ImageNet (Deng et al., 2009) 256×256, is based on the pretrained pixel-space diffusion model provided by Guided-Diffusion (Dhariwal & Nichol, 2021). The order $p$ for student solver DLMS is set to 3. The number of interpolation time steps $M$ is set to 3. We conduct solver distillation on 5k trajectories with default guidance scale 2.0. The distillation times are approximately 6×NFE mins on 8 NVIDIA V100 GPUs. We measure sample quality using the FID score calculated on 10k generated images.

**Stable-Diffusion on MS-COCO prompts.** The text-to-image sampling on MS-COCO (2014) (Lin et al., 2014) 512×512, is based on the pretrained latent-space diffusion model provided by Stable-Diffusion v1.5 (Rombach et al., 2022). The order $p$ for student solver DLMS is set to 2. The number of interpolation time steps $M$ is set to 1. We conduct solver distillation on 5k trajectories with default guidance scale 7.5. The distillation times are approximately 9×NFE mins on 8 NVIDIA V100 GPUs. We measure sample quality using the FID score calculated on 30k generated images generated by 30k prompts from the MS-COCO validation set.

## C   FID RESULTS FOR DLMS AND HANDCRAFTED SOLVERS.

Table 5: Comparison of FID↓ scores between DLMS and handcrafted solvers.

| Dataset | Solver | NFE | | | | | | |
|---|---|---|---|---|---|---|---|---|
| | | 4 | 5 | 6 | 7 | 8 | 9 | 10 |
| **CIFAR10** | DEIS | 25.66 | 14.39 | 9.40 | 6.94 | 5.55 | 4.68 | 4.09 |
| | DPM-Solver++ | 46.52 | 24.97 | 11.99 | 6.74 | 4.54 | 3.42 | 3.00 |
| | UniPC | 45.20 | 23.98 | 11.14 | 5.83 | 3.99 | 3.21 | 2.89 |
| | **DLMS** | **4.52** | **3.23** | **2.81** | **2.53** | **2.43** | **2.37** | **2.24** |
| **FFHQ** | DEIS | 28.31 | 17.36 | 12.25 | 9.37 | 7.59 | 6.39 | 5.56 |
| | DPM-Solver++ | 45.95 | 22.51 | 13.74 | 8.44 | 6.04 | 4.77 | 4.12 |
| | UniPC | 44.78 | 21.40 | 12.85 | 7.44 | 5.50 | 4.47 | **3.84** |
| | **DLMS** | **9.63** | **6.85** | **5.82** | **5.16** | **4.81** | **4.23** | 4.12 |
| **ImageNet-64** | DEIS | 23.53 | 14.75 | 12.57 | 8.20 | 6.84 | 5.97 | 5.34 |
| | DPM-Solver++ | 56.63 | 25.49 | 15.06 | 10.14 | 7.84 | 6.48 | 5.67 |
| | UniPC | 55.63 | 24.36 | 14.30 | 9.57 | 7.52 | 6.34 | 5.53 |
| | **DLMS** | **10.07** | **7.16** | **7.08** | **6.31** | **5.93** | **4.57** | **4.30** |
| **LSUN-Bedroom** | DEIS | 12.89 | 6.83 | 5.35 | 4.78 | 4.43 | 4.25 | 4.05 |
| | DPM-Solver++ | 48.49 | 18.44 | 8.39 | 5.18 | 4.12 | 3.77 | 3.60 |
| | UniPC | 39.66 | 13.76 | 6.46 | 4.52 | 3.96 | 3.72 | 3.56 |
| | **DLMS** | **8.20** | **5.44** | **4.44** | **3.99** | **3.89** | **3.70** | **3.50** |
| **ImageNet** | DPM-Solver++ | 26.30 | 17.08 | 13.06 | 11.08 | 9.95 | 9.25 | 8.72 |
| | UniPC | 24.99 | 15.71 | 11.95 | 10.24 | 9.36 | **8.82** | 8.45 |
| | **DLMS** | **19.74** | **13.83** | **10.90** | **9.66** | **9.07** | 8.89 | **8.22** |
| **MS-COCO** | DEIS | - | 15.16 | 13.95 | 13.61 | 13.58 | 13.46 | 13.38 |
| | DPM-Solver++ | - | 17.40 | 15.78 | 15.08 | 14.77 | 14.43 | 14.18 |
| | UniPC | - | 17.33 | 15.70 | 15.06 | 14.74 | 14.44 | 14.28 |
| | **DLMS** | - | **12.52** | **11.87** | **11.65** | **11.63** | **11.52** | **11.45** |

## D   MORE QUALITATIVE RESULTS

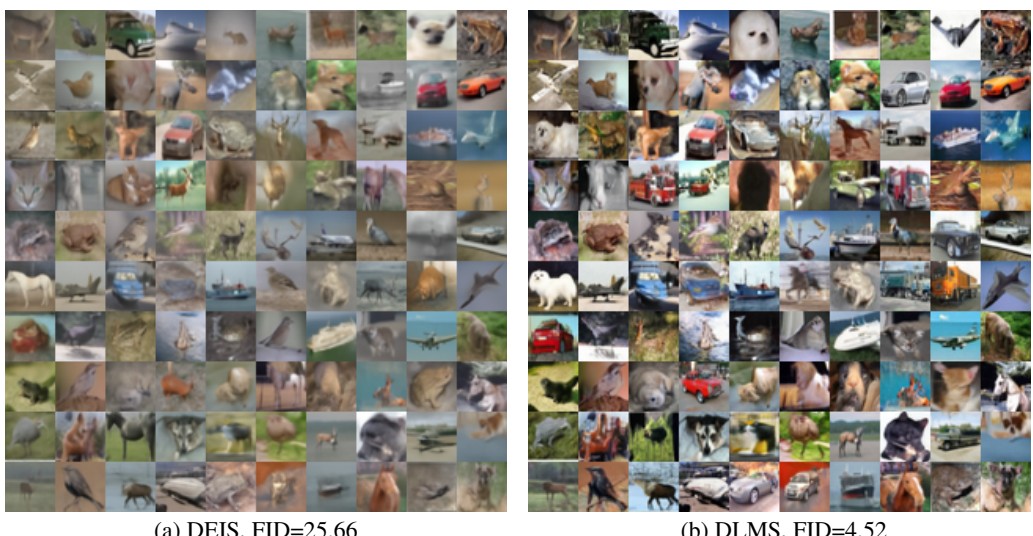

(a) DEIS, FID=25.66                    (b) DLMS, FID=4.52

Figure 5: Uncurated samples on CIFAR10 $32 \times 32$ with 4 NFE.

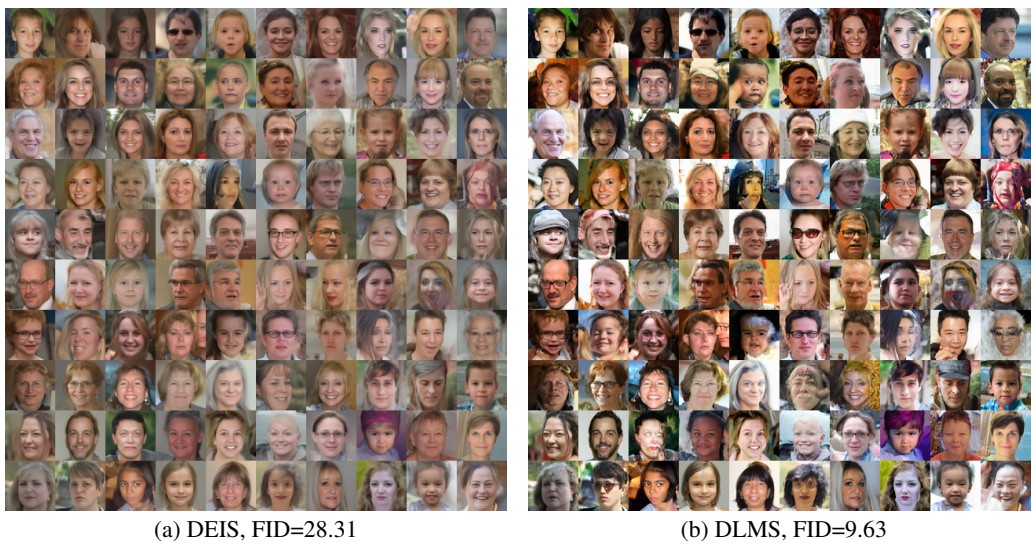

(a) DEIS, FID=28.31                    (b) DLMS, FID=9.63

Figure 6: Uncurated samples on FFHQ $64 \times 64$ with 4 NFE.

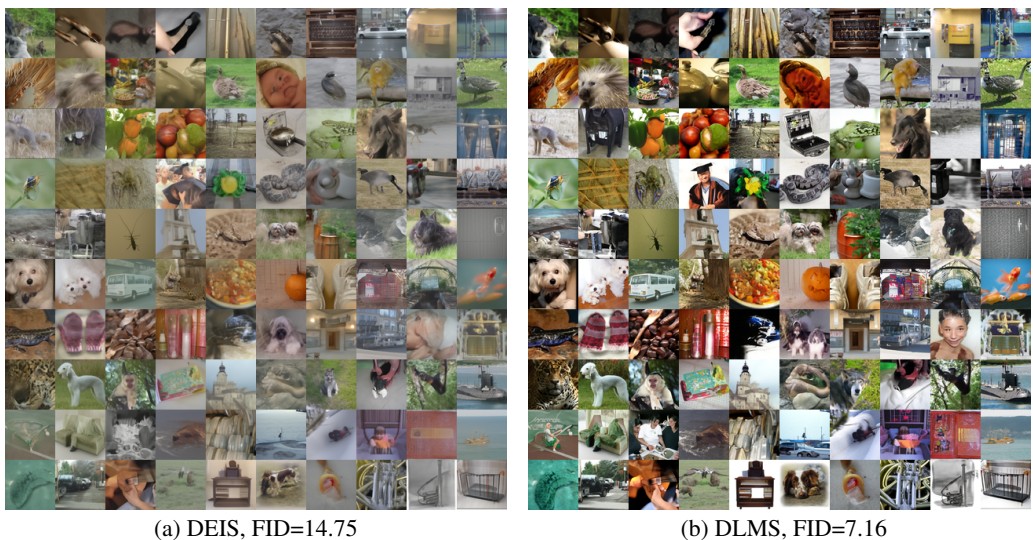

(a) DEIS, FID=14.75          (b) DLMS, FID=7.16

Figure 7: Uncurated samples on ImageNet-64 $64 \times 64$ with 5 NFE.

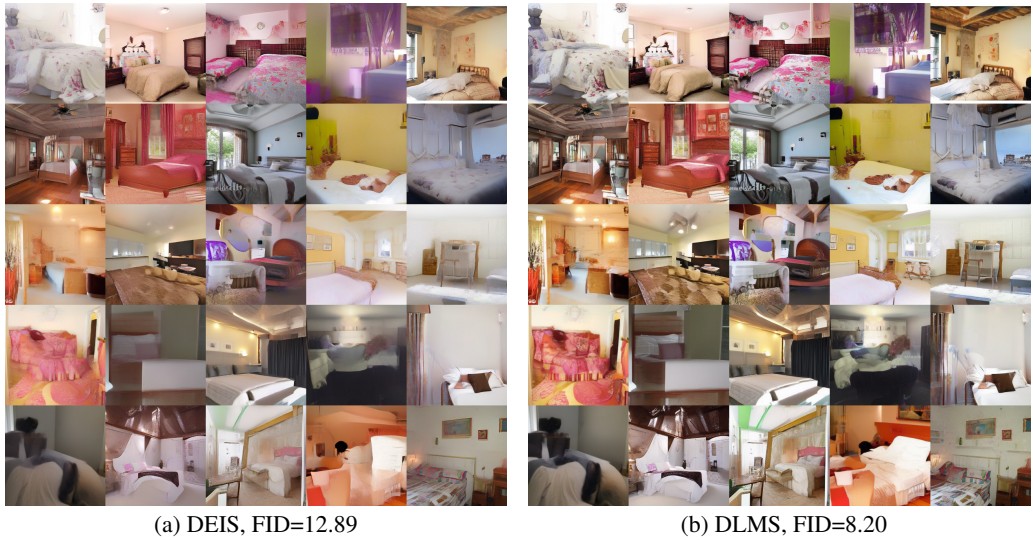

(a) DEIS, FID=12.89          (b) DLMS, FID=8.20

Figure 8: Uncurated samples on LSUN-Bedroom $256 \times 256$ with 4 NFE.

Table 6: Additional samples of Stable-Diffusion (Rombach et al., 2022) with a classifier-free guidance scale 7.5, using only 10 NFE and selected text prompts.

| Text Prompts | DPM-Solver++
(FID=14.18) | DEIS
(FID=13.38) | DLMS
(FID=11.45) |
|---|---|---|---|
| *"a sandwich on wheat bread sits on a plate"* | | | |
| *"three big elephants walking across a wide river"* | | | |
| *"Air force jet in a take off position above the tree line."* | | | |
| *"An empty bench next to a potted tree up against a brick wall."* | | | |
| *"A star hangs upon a canopied bed in a bedroom."* | | | |

