# OpenReview forum: "Linear Multistep Solver Distillation for Fast Sampling of Diffusion Models"
_ICLR.cc/2025/Conference — ICLR 2025 Poster_

### Official Review · Reviewer_Dd8U · 2024-10-29

**Soundness:** 3
**Presentation:** 2
**Contribution:** 2
**Rating:** 6
**Confidence:** 3

**Summary:**

This paper considers diffusion distillation for  a student linear multistep solver from a teach linear multistep solver with more timesteps.  In particular,  the authors propose to train a lightweight neural network to predict the coefficients, time step schedules, and time scaling factors of the student linear multistep solver. The cost function when training the lightweight neural network is taken as the mean squared distance of the difference of diffusion states produced by the student and the teacher linear multipstep solver, respectively. Experiments on FID shows the effectiveness of the new method.

**Strengths:**

The main strength of the paper is that the authors propose to train a lightweight neural network that produce not only the coefficients of the student linear multistep solver but also the timesteps and the scaling factors.  This is based on the assumption that for different ODE trajectories, the optimal coefficients, timesteps and scaling factors are different.

**Weaknesses:**

(0) One weakness is that a small neural network is required to be trained for each particular pre-trained model. Note that not every university or research institute has 8 A00 or H100 GPUs for conducting the training process.

(1) The literature is not thorough. This work is closely related to a recent paper [1], which is not mentioned at all. The work of [1] considers computing the optimal coefficients of a student linear multistep solver per timestep by solving a quadratic optimization problem. The computational complexity of [1] is negligible as the quadratic optimization problem takes a closed-form solution. The authors should include the performance of [1] in their work.

(2) One thing that is not clear to me is results for the two experiments of Latent-Diffusion on LSUN-Bedroom and Stable-Diffusion on MS-COCO prompts, where the number of interpolation timesteps M=1. I would think that the teacher ODE solver with two times of the number of times perform betters than the student ODE solver.  Is it the case? If not, explain why.

(3) I would think that in general, the higher the M value, the better FID score of the student ODE solver. So why in different experimental setups, M were chosen differently? Would higher M value in some cases lead to poor performance? If so, explain why and include the results in the revision.

(4) Typo:  "can be carry out"

[1] Guoqiang Zhang, Kenta Niwa, W. Bastiaan Kleijn, "On Accelerating Diffusion-Based Sampling Processes via Improved Integration Approximation, ICLR, 2024.

**Questions:**

please check my responses above.

**Details Of Ethics Concerns:**

n.a.

---

> ### Author Response · Authors · 2024-11-14
> **Response to ReviewerDd8U**
>
> Dear reviewer Dd8U,
>
> Thank you very much for your review. Here are our responses:
>
> > W0: A small designer network is required to be trained for each pre-trained models. Not every institute has GPUs for the training process.
>
> If you just want to use DLMS on a public model, you can directly download the pre-trained designer network without the need to retrain it, similar to DEIS, DPM-Solver-v3, and [1], which only require pre-computed values to be downloaded beforehand.
>
> However, if you wish to try training the designer network, the cost of our method is not considered unacceptably high. As shown in Table 2, with only one V100 GPU, similar works like UFS and DPM-Solver-v3 require several days to train on the MS-COCO dataset. In contrast, training the designer network can be completed within half a day, allowing more researchers to delve deeper into our method. Additionally, in situations with more limited resources, the cost of our method, apart from text-to-image generation tasks, is very minimal. While there are works with even lower costs, such as [1], it should also be noted that in our work, these costs bring significant benefits.
>
> > W1: This work is closely related to a recent paper [1], which is not mentioned at all.
>
> Thank you for introducing us to this excellent work. [1] proposed a method that utilizes local mean squared error (MSE) to refine existing solvers, which incurs minimal cost but yields limited effectiveness. As we have pointed out in our paper, the choice of time steps can greatly impact the performance of the solvers, with only limited improvements for coefficient optimization. However, the dynamic time steps imply that we need to repeatedly approximate the GT trajectory, incurring some more overheads. Nevertheless,  these overheads also bring significant returns. For instance, on CIFAR10, DLMS achieved an FID of 4.52 with 4 NFEs, outperforming the FID of 6.1 with 10 NFEs achieved by IIA-IPNDM [1]. On MS-COCO, DLMS achieved an FID of 12.52 with 5 NFEs of SDv1.5, surpassing the 12.77 achieved by IIA-DPM-Solver with 20 NFEs of SDv2.
>
> Additionally, our method is data-free, while [1] relies on image datasets. Despite this, we believe that the quadratic optimization method used in [1] may further reduce costs for our method. We would be more than happy to reference that article.
>
> > W2&3: Would higher $M$ value in some cases lead to poor performance?
>
> In fact, the fitting capacity of student solvers is limited. When the teacher solver is significantly superior to the student, side effects can arise due to fitting difficulties. For models in the latent space, setting $M=1$, meaning the teacher has about twice NFEs of the student, yields the best results. The table below illustrates the effects of increasing $M$ on the LSUN-Bedroom dataset.
>
> | NFE | 4 | 6 | 8 | 10 |
> | -- | -- | -- | -- | -- |
> | M=1 | **8.20** | **4.44** | **3.89** | **3.50** |
> | M=2 | 8.86 | 4.56 | 4.51 | 3.69 |
> | M=3 | 11.07 | 4.88 | 4.54 | 3.81 |
>
> Sincerely,
>
> Authors

---

> > ### Comment · Reviewer_Dd8U · 2024-11-27
> >
> > I think the revision addressed most of my concerns. Therefore, we raised my score to 6.

---

### Official Review · Reviewer_23f3 · 2024-11-02

**Soundness:** 4
**Presentation:** 3
**Contribution:** 3
**Rating:** 6
**Confidence:** 4

**Summary:**

This paper proposes DLMS, a flexible solver framework that incorporates the combination of previous model outputs, timestep schedule and timestep scaling factor. The authors further introduce a light weight designer network to dynamically decide the solver strategies for each single trajectory. Experimental results demonstrate DLMS achieves notable improvements over existing solvers and offers faster optimization than search-based methods.

**Strengths:**

1. DLMS offers a flexible framework for diffusion solver, unifying existing methods.
2. DLMS uses dynamic solving strategies for different ODE trajectories, enhacing the potential for diffusion solver.
3. Experimental results demonstrate significant performance improvements compared to existing solvers, and the designer network's training cost is more efficient than that of search-based solvers.

**Weaknesses:**

1. Algorithm 1 implies that each NFE configuration may require independent designer networks, which could limit flexibility in the NFE-performance trade-off. If this is not the case, how does the designer network ensure that $t_N$ becomes a reasonably small when the step count reaches N?
2. The time scaling factor may introduce input distribution misalignment, so further discussion on the motivation and explanation of this would be beneficial.
3. The designer network currently relies on U-Net intermediate feature. As transformers gain popularity in diffusion models, it is uncertain if this approach is adaptable to such architectures.
4. It would be helpful to illustrate differences in solver design choices provided by the designer network across various ODE trajectories to support the claim that a unifed choice for all trajectories is suboptimal.

**Questions:**

1. What is the relationship between the coefficients predicted by the designer network and those derived from Taylor expansion in previous methods? Could you provide a comparison of these coefficients with those from previous methods?

---

> ### Author Response · Authors · 2024-11-16
> **Response to Reviewer 23f3**
>
> Dear reviewer 23f3,
>
> Thank you very much for your review. Here are our responses:
>
> > W1:  Algorithm 1 implies that each NFE configuration may require independent designer networks, which could limit flexibility in the NFE-performance trade-off.  How does the designer network ensure that $t_N$ becomes a reasonably small?
>
> The designer network is trained individually for each NFEs. A related question arises: could we attempt training a universal designer network? While this is clearly feasible, it also implies more complex inputs and a multiplied parameter count. Repeatedly invoking a more complex designer network during sampling would directly raise the sampling costs. Therefore, it seems more reasonable to opt for multiple independent designer networks. Methods like USF and AMED also incorporate additional networks, all of which use separate networks.
>
> When the *step_index* is $N-1$, we constrain the network to directly output $t_N=0$ (which is actually a positive value close to 0). We also enforce that the output $t_n$ must be less than the input $t_{n-1}$.
>
> > W2: The time scaling factor may introduce input distribution misalignment, so further discussion on the motivation and explanation of this would be beneficial.
>
> Introducing errors when solving ODEs is inevitable, implying that the estimated value $X_t^S$ contains more noise compared to the groundturth value $X_t$, which is known as exposure bias. In such cases, a time scale slightly greater than 1 is appropriate.
> Meanwhile, during the solving process from $t_{n-1}$ to $t_n$, what we need to estimate is the integral from $t_{n-1}$ to $t_n$, where a time scale less than 1 can play a role similar to the mean value theorem.
>
> The principle behind the role of the time scale is complex. Taking the experiment with 7 NFEs on Stable Diffusion as an example, the sequence $\{s_n\}$ is $1.013, 1.001, 0.999, 0.935, 0.888, 0.838, 0.880. $ For large $t$, the time scales will be slightly greater than 1, while for $t$ close to 0, the time scales will be less than 1.
>
> > W3: The designer network currently relies on U-Net intermediate feature. As transformers gain popularity in diffusion models, it is uncertain if this approach is adaptable to such architectures.
>
> In principle, we aim to extract a particular layer of features from the backbone of the diffusion model to reduce the dimensionality of the features. While the bottleneck layer is a natural choice, it is not mandatory. Additionally, it is worth noting that even without relying on feature information, our method can still significantly improve solving performance through adaptive coefficients, time steps, and time scales. It is also worth mentioning that the U-Net architecture seems to have regained a prominent position in diffusion models [1][2].
>
> > W4:  It would be helpful to illustrate differences in solver design choices provided by the designer network across various ODE trajectories to support the claim that a unified choice for all trajectories is suboptimal.
>
> The differences in solving trajectories primarily show in the time steps. Taking the case of 10 NFEs on the Stable Diffusion model as an example, our paper notes a basic linear relationship in the learned half-log-snr at each step. However, in practice, there are some variations in the time step across different trajectories. The table below lists the mean and standard deviation of half-log-snr at each step for reference.
>
> | Step Index | 0 | 1 | 2 | 3 | 4 | 5 | 6 | 7 | 8 | 9 |
> | -- | -- | -- | -- | -- | -- | -- | -- | -- | -- | -- |
> | Mean | -2.488 | -2.064 | -1.675 | -1.225 | -0.733 | -0.308 | 0.107 | 0.531 | 1.007 | 1.438 |
> | Standard Deviation | .000 | .013 | .013 | .012 | .013 | .018 | .019 | .022 | .021 | .022 |
>
> > Q1: What is the relationship between the coefficients predicted by the designer network and those derived from Taylor expansion in previous methods?
>
> We believe that Taylor expansion is a compromise in the absence of real trajectory values. DLMS, as a data-driven method, does not require excessive emphasis on interpretability. The table below shows a comparison of the inference coefficients for the 5th step at 10 NFEs on the LSUN-Bedroom dataset. It can be observed that DEIS is quite close to the DLMS method, and in many cases, it is also the best-performing handcrafted solver.
>
> | Solver | $a_1$ | $a_2$ | $a_3$ |
> | -- | -- | -- | -- |
> | PLMS | 1.91 | -1.33 | 0.41 |
> | DPM-Solver++ | 1.67 | -0.90 | 0.23 |
> | DEIS | 1.83 | -1.21 | 0.38 |
> | DLMS | 1.78 | -1.06 | 0.30 |
>
> Sincerely,
>
> Authors
>
> References
>
> [1] Tian Y, Tu Z, Chen H, et al. U-DiTs: Downsample Tokens in U-Shaped Diffusion Transformers. NIPS 2024.
>
> [2] Bao F, Nie S, Xue K, et al. All are worth words: A vit backbone for diffusion models. CVPR 2023.

---

> > ### Comment · Reviewer_23f3 · 2024-12-03
> > **Thank you for your rebuttal**
> >
> > Thank you for your detailed rebuttal, which has addressed my concerns. I will maintain my score and recommend acceptance.

---

### Official Review · Reviewer_AsEn · 2024-11-02

**Soundness:** 3
**Presentation:** 3
**Contribution:** 3
**Rating:** 8
**Confidence:** 3

**Summary:**

The paper proposes a method for learning a trajectory-specific solver for diffusion models. The method suggest to use a small network to predict the best time step size and coefficients of a linear multistep at each step of the solver. The method is tested on a number of image generation tasks.

**Strengths:**

1. The idea to learn a trajectory-specific solver is novel and interesting.
2. The method shows good results on number of benchmark datasets.
3. The method is compared to a number of diffusion dedicated solvers.

**Weaknesses:**

1. The authors make use of what they call "bottleneck feature" without any explanation what are those features, only referencing the relevant paper. The paper should be self contained and the authors should make an effort to give even brief explanation about these features.
2. The method is not compared to any other solver distillation methods such as [1], [2].
3. Discussion and comparison to model distillation is too minimal.
4. The size of the designer network is not provided.

**Questions:**

1. Does a different designer network needs to be learned for each choice of NFE, multistep order?
2. Is the designer network is somehow constrained such that always $t_N=0$?
3. Could the authors provide the size of the designer network?
4. The designer network is dependent only on $h_{t_{n-1}}$ or previous times as well?


[1] Zheng, Kaiwen, et al. "Dpm-solver-v3: Improved diffusion ode solver with empirical model statistics." Advances in Neural Information Processing Systems 36 (2023): 55502-55542.

[2] Shaul, Neta, et al. "Bespoke Non-Stationary Solvers for Fast Sampling of Diffusion and Flow Models." arXiv preprint arXiv:2403.01329 (2024).

---

> ### Author Response · Authors · 2024-11-14
> **Response to Reviewer AsEn**
>
> Dear reviewer AsEn,
>
> Thank you very much for your review. Here are our responses:
>
> > Q1: Does a different designer network needs to be learned for each choice of NFE, multistep order?
>
> Yes, the designer network is trained individually for each NFEs. A related question arises: could we attempt training a universal designer network? While this is clearly feasible, it also implies more complex inputs and a multiplied parameter count. Repeatedly invoking a more complex designer network during sampling would directly raise the sampling costs. Therefore, it seems more reasonable to opt for multiple independent designer networks. Methods like USF and AMED also incorporate additional networks, all of which use separate networks.
>
> > Q2: Is the designer network is somehow constrained such that always $t_N=0$ ?
>
> Yes, when the *step_index* is $N-1$, we constrain the network to directly output $t_N=0$ (which is actually a positive value close to 0). We are willing to share further details about the designer network. We also enforce that the output $t_n$ must be less than the input $t_{n-1}$, which is essential in some experiments. Moreover, for positive outputs such as time step $t_n$ and time scale $s_n$, we actually apply exponential transformations.
>
> > Q3 Could the authors provide the size of the designer network?
>
> In line 372, we highlighted that the designer network comprises a two-layer MLP with a total parameter count of only 9k. This is negligible compared to the parameter count of diffusion models (e.g., SDv1.5 with a parameter count of 680M). Perhaps we should mention this earlier in our article. Thank you for your suggestion.
>
> > Q4: The designer network is dependent only on $h_{t_{n-1}}$ or previous times as well?
>
> Due to the necessity for the designer network to be very lightweight, its input needs to have a low dimensionality. Therefore, we chose to extract features from the bottleneck layer of the U-Net and then compress them into a vector of only 64 dimensions through average pooling. For the Text-to-Image model, the bottleneck features encompass image, text, and time information, ensuring that the input to the designer network is concise yet informative. We only utilize the most relevant $h_{t_{n-1}}$ and discard previous features to simplify the input and save on space costs.
>
> > W2: The method is not compared to any other solver distillation methods such as DPM-Solver-v3, Bespoke Solver.
>
> In Tables 1 and 2, our DLMS method demonstrates significant superiority in performance and training costs compared to DPM-Solver-v3. DPM-Solver-v3 performs impressively in scenarios like Latent Diffusion, which are insensitive to time steps. In fact, we could use DPM-Solver-v3 to initialize distillation training, but we prefer our method not to rely on a similar approach.
>
> Bespoke solver is also a type of solver distillation method, but it only uses the final output to adjust hyperparameters, with a deep and complex computation graph that hinders effective hyperparameter tuning. We actively use intermediate values in the trajectory and judiciously stop unnecessary gradient backpropagation, greatly enhancing the effectiveness of solver distillation method.
>  In the publicly available model EDM  trained on CIFAR10, our method achieved an FID of 2.53 with 7 NFEs, outperforming their result of 2.75 with 20 NFEs. Nevertheless, we are more than willing to cite this highly relevant article.
>
> > W3: Discussion and comparison to model distillation is too minimal.
>
> We agree that a more detailed introduction to model distillation is necessary, and we intended to add relevant content.
>
> Sincerely,
>
> Authors

---

> > ### Comment · Reviewer_AsEn · 2024-11-25
> >
> > I want thank the authors for their response, they have addressed my questions/concerns, and I will raise my rating.

---

### Official Review · Reviewer_V3hM · 2024-11-04

**Soundness:** 3
**Presentation:** 3
**Contribution:** 3
**Rating:** 8
**Confidence:** 3

**Summary:**

This paper proposes a Distilled Linear Multistep Solver (DLMS) to learn a faster sampler for diffusion models, requiring fewer function evaluations. The distillation approach is to train a solver that minimizes the Euclidean distance between its trajectory and a teacher solver’s trajectory. DMLS can be trained faster than previous reinforcement learning-based approaches to solver distillation. Experimental results in image generation using unconditional, conditional, latent space, and pixel space diffusion show improved FID scores compared to existing methods, especially in low NFE settings.

**Strengths:**

The strengths of the paper are its well-motivated method and a wide range of experiments.
- The approach can be initialized using existing solvers.
- Unlike model distillation, solver distillation like DLMS can be used for downstream tasks like image restoration.
- The proposed method is simpler than using a reinforcement learning-based approach.
- The method is evaluated in multiple contexts (unconditional, conditional, latent space, and pixel space diffusion) and on multiple datasets with convincing results.
- The paper is overall well-written.

**Weaknesses:**

The weaknesses of the paper include its figure and table captions and reliance on FID as the sole quantitative metric.
- In general, the figures and table are not self-contained. Figure 4, for example, could be made more interpretable by describing the significance of dashed lines.
- FID scores are the only quantitative metric used to evaluate the method. Quantifying the quality of generated images is challenging, so adding multiple metrics like IS or CMMD [1] would increase confidence in the results.
- In the introduction (line 047) it is argued that distillation is expensive, requiring multiple GPU days of training. The reported training times for DLMS are still more than ten hours, so these methods could still be compared, if not to better understand their respective strengths and weaknesses.
- The time comparisons in Table 2 are hard to draw conclusions from considering the reported times are compared to those from previous papers that were run on different systems with different GPUs and software environments. It is stated that this is due to limited code availability. If they are possible to obtain, FLOP counts (or an estimate of them) would be more comparable.

**References**

[1] Jayasumana, S., Ramalingam, S., Veit, A., Glasner, D., Chakrabarti, A. and Kumar, S. “Rethinking FID: Towards a Better Evaluation Metric for Image Generation”, CVPR 2024

**Questions:**

- On line 398 it is reported that the distillation time of DMLS is approximately 1.5 * 8 = 12 hours for stable diffusion, while the abstract claims that the framework “has the ability to complete a solver search for Stable-Diffusion in less than 10 total GPU hours”. Is this difference due to a rounding error or do these claims refer to different times? Clarify this discrepancy and ensure consistency between the abstract and results.

**Minor suggestions that do not individually affect the score**
- Line 129: “can be carry out” -> “can be carried out”.
- Line 189: Remove “precious”.
- Line 263: Introduce the strop gradient operation.
- Line 284: Reformulate.
- Line 292: “PLMS(iPNDM)” -> “PLMS (iPNDM)”.
- Line 340: “AMED-Plugin(Zhou et al., 2024)” -> “AMED-Plugin (Zhou et al., 2024)”.
- Line 363: Specify “various aspects” and “as well as the ablation…” -> “as well as ablation…”.
- Line 394: “MS-COCO(2014)” -> “MS-COCO (2014)”.
- Line 485: “Handcrafted(best)” -> “Handcrafted (best)”.

---

> ### Author Response · Authors · 2024-11-20
> **Response to Reviewer V3hM**
>
> Dear reviewer V3hM,
>
> Thank you very much for your review. Here are our responses:
>
> > Q1: On line 398 it is reported that the distillation time of DMLS is approximately 1.5 * 8 = 12 hours for stable diffusion, while the abstract claims that the framework “has the ability to complete a solver search for Stable-Diffusion in less than 10 total GPU hours”. Is this difference due to a rounding error or do these claims refer to different times? Clarify this discrepancy and ensure consistency between the abstract and results.
>
> The primary time consumption is in generating the teacher trajectory, which is roughly proportional to the number of NFEs. In the Stable Diffusion experiments, 5-10 NFEs actually correspond to 0.75 h to 1.5 h. We will revise to a more specific description, such as "about $9 \times NFEs$ mins". We believe that precise statements are crucial, and we sincerely appreciate you pointing this out.
>
> > W2: Quantifying the quality of generated images is challenging, so adding multiple metrics like IS or CMMD [1] would increase confidence in the results.
>
> To be frank, while DLMS significantly outperforms handcrafted solvers in terms of FID and visual presentation, CMMD lags behind. It took us several days to understand why. We discovered that, compared to images rich in texture details, CMMD tends to prefer images with simple compositions, clean visuals, and clear semantics.
>
> After comparing CMMD values between DPM-Solver++ at 10 NFE and 20 NFE, we were surprised to find that the CMMD value of **0.578** at **10** NFE is better than the value of **0.583** at **20** NFE. Then everything makes sense now, because our DLMS at 10 NFE emulates the behavior of DPM-Solver++ at 20 NFE as its teacher.
>
> The CMMD scores of more powerful generative models like SDXL and PixArt are significantly worse than the CMMD of SDv1.5. The authors of CMMD attribute this to the COCO dataset. However, we believe that adopting MMD is reasonable, with the issue possibly lying in the CLIP model's excessive emphasis on semantic information.
>
> > W3: In the introduction, it is argued that distillation is expensive, requiring multiple GPU days of training. The reported training times for DLMS are still more than ten hours, so these methods could still be compared, if not to better understand their respective strengths and weaknesses.
>
> Model distillation typically requires thousands of GPU hours. The intuitive difference in magnitude can better help highlight our contribution. Thank you for your suggestion. Additionally, we plan to provide more specific introductions about downstream tasks, including guidance methods and post-processing techniques. This will help understand why solvers can be more flexibly applied to downstream tasks.
>
> > W1 & Minor suggestions:  In general, the figures and table are not self-contained. Figure 4, for example, could be made more interpretable by describing the significance of dashed lines.
>
> Thank you once again for these helpful writing suggestions. We will make corrections accordingly.
>
> Sincerely,
>
> Authors

---

> > ### Comment · Reviewer_V3hM · 2024-11-25
> >
> > Thank you for the response.
> >
> > **Q1**: Ok, that is reasonable (although I cannot see the change in the current revision).
> >
> > **W2**: Interesting. Indeed, a metric that prefers samples generated with a coarser solver is not a good metric for evaluating solvers. Regardless, I am willing to look past the reliance on FID as the only metric because it most likely gives a correct indication of the sample quality and is a de facto standard metric for measuring generation quality at the time of writing, and this paper does not set out to change that.
> >
> > **W1 + W3**: These have been improved in the revised paper.
> >
> > I have no further questions and will maintain the score.

---

### Author Response · Authors · 2024-11-20
**Revision Summary**

Dear AC and reviewers,

We have revised the paper according to the reviews. The modifications are as follows:

 1. We find that adding constraint $\sum_{k=1}^p a_{k}=1$ ensures the unbiasedness of the estimation. This significantly improves the performance of the DLMS.  We will update new results gradually.
The table below shows the new results on Stable Diffusion v1.5.
|NFE | 5 | 6 | 7 | 8 | 9 | 10 |
|--|--|--|--|--|--|--|
| DEIS | 15.16 | 13.95 | 13.61 | 13.58  |13.46 | 13.38 |
| DLMS w/o constraint | 15.69 | 13.54 | 13.20  | 13.07 | 12.69 | 11.72 |
| DLMS w constraint | **12.52**  | **11.87** | **11.65** | **11.63** | **11.52** | **11.45** |

 2. A more detailed discussion on the limitations of model distillation in image processing problems has been added as Sec. 2.2.

 3. Added two related recent works [1] [2] in introduction and related works. And added the comparison with Bespoke solver in Sec 3.4.

 4. A sensitivity analysis of number of interpolation time steps $M$ is added in Sec. 4.3.

 5. Add more description to the tables and figures.

 6. The typos mentioned have been corrected.

Sincerely,

Authors

References

[1] Shaul, Neta, et al. "Bespoke Non-Stationary Solvers for Fast Sampling of Diffusion and Flow Models.", ICML, 2024.

[2] Guoqiang Zhang, Kenta Niwa, W. Bastiaan Kleijn, "On Accelerating Diffusion-Based Sampling Processes via Improved Integration Approximation, ICLR, 2024.

---

### Meta-Review · Area_Chair_Lvom · 2024-12-18

**Metareview:**

This paper introduces DLMS, a novel method that trains a lightweight neural network to generate optimal coefficients, timesteps, and scaling factors for a linear multistep ODE solver, tailored to specific trajectories. This approach offers a flexible framework unifying existing diffusion solvers and demonstrates significant performance improvements over them.  Key strengths highlighted by reviewers include:

Reviewers agreed that the proposed approach for learning trajectory-specific solvers demonstrates strong empirical results on benchmark datasets across diverse diffusion contexts and is effective in lowering training cost compared to search-based solvers.

**Additional Comments On Reviewer Discussion:**

- Updated empirical results
- Additional discussion on limitations
- Additional discussion on related works
- Sensitivity analysis
- More discussion on empirical results

See "Revision Summary" by the authors.

---

### Decision · Program_Chairs · 2025-01-22

Accept (Poster)